# In Silico Analysis and Development of the Secretory Expression of D-Psicose-3-Epimerase in *Escherichia coli*

**DOI:** 10.3390/microorganisms12081574

**Published:** 2024-08-01

**Authors:** Nisit Watthanasakphuban, Boontiwa Ninchan, Phitsanu Pinmanee, Kittipong Rattanaporn, Suttipun Keawsompong

**Affiliations:** 1Department of Biotechnology, Faculty of Agro-Industry, Kasetsart University, Chatuchak, Bangkok 10900, Thailand; faginsw@ku.ac.th (N.W.); fagibin@ku.ac.th (B.N.); kittipong.r@ku.th (K.R.); 2Enzyme Technology Research Team, National Center of Genetic Engineering and Biotechnology (BIOTEC), Pathum Thani 12120, Thailand; phitsanu.pin@biotec.or.th

**Keywords:** D-psicose-3-epimerase, psicose, protein secretion, signal peptide, protein expression

## Abstract

D-psicose-3-epimerase (DPEase), a key enzyme for D-psicose production, has been successfully expressed in *Escherichia coli* with high yield. However, intracellular expression results in high downstream processing costs and greater risk of lipopolysaccharide (LPS) contamination during cell disruption. The secretory expression of DPEase could minimize the number of purification steps and prevent LPS contamination, but achieving the secretion expression of DPEase in *E. coli* is challenging and has not been reported due to certain limitations. This study addresses these challenges by enhancing the secretion of DPEase in *E. coli* through computational predictions and structural analyses. Signal peptide prediction identified PelB as the most effective signal peptide for DPEase localization and enhanced solubility. Supplementary strategies included the addition of 0.1% (*v*/*v*) Triton X-100 to promote protein secretion, resulting in higher extracellular DPEase (0.5 unit/mL). Low-temperature expression (20 °C) mitigated the formation of inclusion bodies, thus enhancing DPEase solubility. Our findings highlight the pivotal role of signal peptide selection in modulating DPEase solubility and activity, offering valuable insights for protein expression and secretion studies, especially for rare sugar production. Ongoing exploration of alternative signal peptides and refinement of secretion strategies promise further enhancement in enzyme secretion efficiency and process safety, paving the way for broader applications in biotechnology.

## 1. Introduction

Rare sugars have been defined by the International Society of Rare Sugars (ISRS) as monosaccharides and derivatives that rarely exist in nature and are different from other monosaccharides, such as D-glucose, D-fructose, D-galactose, D-xylose, D-ribose, and L-arabinose, which are commonly found in a variety of stereoisomer forms in nature [1,2]. Some rare sugars have gained more attention because of their health beneficial characteristics, such as antioxidant activity, anti-inflammatory activity, cancer and tumor inhibition, reduction in reactive oxygen species (ROS), and their use as a low- or zero-calorie sweetener [3]. To date, 29 monosaccharides have been classified as rare sugars, including hexoses and pentoses, such as D-arabinose, D-psicose, D-tagatose, L-glucose, L-fructose, L-galactose, L-mannose, D-talose, D-xylulose, D-xylitol, and D and L isomers of allose, gulose, idose, lyxose, ribulose, and sorbose [4].

D-psicose (D-allulose, D-ribo-2-hexulose), a C3 epimer of D-fructose, is one of the rare sugars that is found naturally only in very small amounts of about 0.38 and 0.29 mg/g in raisins and dried figs ([5,6]. D-psicose has 70% sweetness compared to sucrose but has a very low energy level of only 0.007 kcal/g and is poorly absorbed through the digestive tract [5]. D-psicose is nontoxic to cells [7,8] and has been granted Generally Recognized As Safe (GRAS) status by the United States Food and Drug Administration as well as being approved as a food ingredient, making it more attractive for food and functional ingredient applications. In addition, the health benefits associated with D-psicose include reduced blood glucose level and anti-inflammation [9], reduced visceral fat mass [10], and hypoglycemic properties and therapeutic effects on type 2 diabetes [11].

D-psicose is difficult to synthesize chemically, so the Izumering strategy was applied for the conversion of D-fructose to D-psicose using an enzyme called D-psicose-3- epimerase (DPEase) [1]. This enzyme is present in some microorganisms, including *Agrobacterium tumefaciens* [12]; however, DPEase production using wild-type microorganisms has a low yield and is not applicable for industrial due to cost inefficiency. Recombinant DNA technology has been applied to improve the enzyme yield, with a resulting greater production yield of DPEase being reported using *Escherichia coli*, *Bacillus subtilis*, and *Corynebacterium glutamicum* [7,13]. Most studies on recombinant DPEase have focused on intracellular enzyme production, which requires complicated downstream processing that has high production costs. The secretion of DPEase has only been reported in engineered *B. subtilis* [14,15,16] with some limitations in the *Bacillus* expression system such as it being the only option suitable for expression vectors, plasmid instability, high protease production and occurrence of misfolded proteins [17]. The secretion of DPEase in *E. coli*, one of the major industrial cell factories with a secretion systems [18], has not yet been successfully constructed and reported. Although OmpA has been reported as a prominent and efficient signal peptide in *E. coli*, recent findings have linked it to virulence in pathogenic Enterobacteriaceae [19,20,21,22,23,24,25]. Consequently, the exploration of alternative signal peptides for DPEase secretion in *E. coli* could reduce the endotoxin contamination concerns from the *E. coli* expression system and support the use of recombinant DPEase for industrial applications.

This study aims to address the challenges and limitations surrounding the secretion of DPEase in *E. coli* through structural analysis. By leveraging a combination of in silico enzyme localization prediction and media composition strategies, we successfully overcame the secretion hurdles associated with DPEase. This finding facilitated the development of an efficient DPEase-secretion system in *E. coli*, representing a significant stride in DPEase cell factory production. Notably, this advancement mitigated endotoxin concerns by eliminating the need for cell membrane disruption and employing filtration techniques, thereby minimizing downstream processing requirements.

## 2. Materials and Methods

### 2.1. Bacterial Strains, Plasmids, and Culture Conditions

The relevant features of expression of the plasmids, primers, and bacterial strains used in this work are listed in Table 1.

*E. coli* Neb5α was used as an intermediate cloning host, along with the *E. coli* BL21(DE3) expression host grown at 37 °C in LB solid medium containing 1.5% (*w*/*v*) agar with agitation at 200 rpm [26,27]. Antibiotics were used at a final concentration of 50 μg/mL of kanamycin (kan) for pET28a-carrying strains.

### 2.2. In Silico Analysis of DPEase and Signal Peptide Prediction for DPEase Localization

The optimized sequence of DPEase was analyzed and the protein structure was elucidated using the refernce DPEase enzyme structure of *A. tumefaciens* CS58 (Uniprot identifier: A9CH28). Both the N-terminal and C-terminal regions of DPEase were examined, and the gap distance was measured prior to signal peptide selection. Signal peptides known for high secretion efficiency in *E. coli* [28] were chosen and fused to the optimized DPEase gene (Appendix A), with a comparison made to DPEase lacking a signal peptide. Amino acid sequences were assessed for solubility of DPEase constructs using SOLUPROT V1 (https://loschmidt.chemi.muni.cz/soluprot/, accessed on 21 March 2024), DPEase localization was predicted using DeepLocPro 1.0 (https://services.healthtech.dtu.dk/services/DeepLocPro-1.0/, accessed on 22 March 2024), and cleavage position and cleavage probability were calculated using SignalP 6.0. The optimal DPEase construct with signal peptides was selected for recombinant plasmid construction, in comparison with the commonly used *E. coli* signal peptide, OmpA.

### 2.3. Construction of Plasmids and Molecular Cloning of DPEase Recombinants

The DPEase encoding gene from *Agrobacterium tumefaciens* C58 (GenBank accession no. AAK88700.1), with a size of 870 bp, was artificially synthesized based on codon optimization using GenScript (Piscataway, NJ, USA). The construction of the pET28a + DPEase, pET28a + OmpA + DPEase, and pET28a + PelB + DPEase expression plasmids was carried out using standard digestion and ligation procedures. The synthesized DPEase gene was ligated into the *Nco*I and *Xho*I cloning sites of pET28a, pET28a + OmpA, and pET28a + PelB, with the ligation mixtures transformed into *E. coli* Neb5α and the *E. coli* BL21(DE3)-competent cells [26,27] using standard *E. coli* heat-shock transformation. The DPEase expression constructs were verified using colony polymerase chain reaction and sequencing with specific primers.

### 2.4. Localization and Expression of DPEase and Preparation of Recombinant Enzymes from Various Compartments

The resulting *E. coli* recombinant strains harboring DPEase plasmid constructs (pET28a + DPEase, pET28a + OmpA + DPEase, pET28a + PelB + DPEase) were streaked onto LB agar containing the appropriate antibiotic. A single colony was picked and inoculated into 5 mL LB broth containing the antibiotic and cultivated at 37 °C for 16 h. Next, 250 μL of overnight culture was transferred into 50 mL LB broth containing the antibiotic and grown at 37 °C at 200 rpm until the OD 600 nm reached 0.4–0.6. Then, isopropyl ß-D-1-thiogalectopyranoside (IPTG) was added into the culture broth to a final concentration of 1 mM, and further incubated at 30 °C for 16 h. After collecting the overnight cultures, the cells and cell-free supernatants were harvested and prepared into 3 fractions: fermentation broth, periplasm fraction, and cell lysate.

Fermentation broth sample: the cell-free supernatants of Psicose, OmpA_Psicose, and PelB_Psicose were collected and concentrated (10-fold) using a 10 kDa cut-off centrifugal filter (Merck, Darmstadt, Germany).

The periplasm fractions were extracted by resuspending the cells with 30 mL spheroplast buffer (30 mM Tris-HCl buffer, pH 8.0, 20% (*w*/*v*) sucrose, and 0.5 mM ethylenediaminetetraacetic acid (EDTA)) and gently shaken for 10 min at room temperature. Then, the spheroplast buffer was removed using centrifugation (10,000 rpm, 10 min, 4 °C), and the cells were resuspended in 30 mL of cold 5 mM MgSo_4_ and gently shaken for 10 min at 4 °C. The extracted periplasm fraction was collected using centrifugation (10,000 rpm, 10 min, 4 °C), according to a previous report [29].

The cell lysates were extracted from the precipitated cells from the previous step after washing twice with 50 mM sodium phosphate buffer, pH 7.5. The cells were resuspended in 5 mL cold lysis buffer (50 mM sodium phosphate buffer, pH 7.5, 0.1% (*v*/*v*) Triton X-100) and sonicated with 60 amplitude and a pulser set at 10 s for 2 min on ice. The cell lysates were harvested using centrifugation (15,000 rpm, 30 min at 4 °C), and cell debris was separated.

The collected fractions were measured for their protein concentration using the Bradford assay (Bradford, 1976) with bovine serum albumin as a standard protein, and the size of the DPEase was identified based on sodium dodecyl sulphatepolyacrylamide gel electrophoresis (SDS-PAGE) with Coomassie Brilliant Blue G-250 staining [26]. Enzyme activity was analyzed using the conversion efficiency of D-fructose to D-Psicose and detected using high-performance liquid chromatography (HPLC) by comparing to D-fructose and D-psicose standard.

### 2.5. Effect of Triton X-100 on the Localization and Expression of DPEase

The *E. coli* recombinant strains harboring DPEase plasmid constructs (pET28a + DPEase, pET28a + OmpA + DPEase, pET28a + PelB + DPEase) were cultured and induced with IPTG following the method described above; however, 0.1% (*v*/*v*) Triton X-100 was supplemented into the LB broth during expression. Then, the overnight cultures were centrifuged, and the cells and cell-free supernatants were harvested and prepared into the 3 fractions and their protein expression and enzyme activity were measured.

### 2.6. Effect of Temperature on DPEase Expression

The effect of DPEase expression was investigated by adjusting the culture conditions after induction to various temperatures. The *E. coli* recombinant strains harboring the DPEase plasmid construct were cultured in 50 mL LB broth containing the antibiotic and grown at 37 °C and 200 rpm until the OD 600 nm reached 0.4–0.6. Then, 1 mM IPTG was added, and each strain was further incubated at four different temperatures (20 °C, 25 °C, 30 °C, and 37 °C) for 16 h. The cell lysate samples were measured for protein and enzyme activity, and the best culture conditions was applied for the PelB_Psicose cultivation. The fermentation broth, periplasm fraction, and cell lysate of PelB_Psicose were collected; the protein concentration and the enzyme activity were analyzed.

### 2.7. Enzyme Assay and Sugar Analysis

Enzyme activity was measured following the protocol described by Kim et al. (2006), with some modifications. DPEase fractions were measured for their enzyme activity after incubation at 4 °C with 1 mM Mn^2+^ in 100 mM Tris-HCl buffer (pH 8.0) for 4 h. The fructose solution (1% (*w*/*v*) fructose in 100 mM Tris-HCl, pH 8.0) was added into DPEase fractions using a 1:1 ratio, and the reaction was performed at 50 °C for 10 min and stopped by boiling for 10 min.

The D-psicose and fructose concentrations were measured using an HPLC (RID-10A; Shimadzu Corporation; Kyoto, Japan) fitted with a refractive index detector (RI) at 40 °C and a VertiSep^TM^ Sugar CMP column (7.8 × 300 mm; Vertical Chromatography Co. Ltd.; Nonthaburi, Thailand). Twenty microliters of each sample, was injected and the elution was performed at 80 °C using water at a flow rate of 0.4 mL/min. One unit of DPEase activity was defined as the amount of enzyme producing 1 μmol of D-psicose per min at 50 °C and pH 8.0 [12].

The specific activity of DPEase (unit/mg) was calculated by dividing DPEase activity (unit/mL) by protein concentration (mg/mL).

## 3. Results

### 3.1. In Silico Analysis of Signal Peptide for DPEase Localization

The 25 signal peptides (sp) selected from their high secretion efficiency (greater than 85%) in *E. coli* from a previous report [28] were fused to the optimized DPEase encoding gene (sp_DPEase). The predicted secretory pathways for each signal peptide were determined using SignalIP6.0 and are summarized in Table 2. Most selected signal peptides were predicted to utilize the Sec-dependent pathway with high probability, except for DPEase fused with the ampC signal peptide (ampC_DPEase), which showed a lower probability and was predicted to employ secretion via both the Sec-pathway (Sec/SPI) and a lipoprotein signal peptide (Sec/SPII) (Table 2). When the DPEase encoding gene was not fused with a signal peptide (DPEase), no secretory pathway was detected (Other).

The localization of the DPEase encoding gene without signal peptide was predicted to be intracellular with a probability of 0.9936 (DPEase) (Table 3). However, fusion with any of the selected signal peptides resulted in a shift in the localization of DPEase from intracellular to the periplasm. The three highest probabilities were observed when the DPEase sequence was fused with PelB, malE, and ppiA signal peptides.

Additionally, the addition of a signal peptide remarkably influenced the solubility of DPEase. The highest solubility of 0.919 was predicted for the DPEase sequence without a signal peptide (DPEase) (Table 3), whereas fusion with any signal peptide resulted in a reduction in the solubility of DPEase. Among the DPEase sequences with PelB signal peptide fusion, PelB_DPEase exhibited the highest solubility, with a value of 0.89, while only 0.79 was predicted when OmpA was fused to DPEase (OmpA_DPEase).

Based on the in silico analysis, the PelB signal peptide, which exhibited the highest solubility and localization efficiency, was chosen for constructing the DPEase secretion construct. Additionally, the widely used OmpA signal peptide was selected to facilitate a comparison of secretion efficiency for in vitro analysis, thereby validating the results obtained from the in silico analysis.

### 3.2. The DPEase Structure Analysis

According to the in silico analysis of signal peptides for DPEase solubility, fusion with any signal peptide reduced the solubility of DPEase. Therefore, the DPEase structure was analyzed for the possibility or limitation of secretion in *E. coli*. The predicted structure of DPEase, using the amino acid sequence fused with the PelB signal peptide and 6xHis (PelB_DPEase, Appendix A), showed a similar DPEase structure to what has been reported [30]. Thus, the references DPEase structure from *A. tumefaciens* (Uniprot identifier: A9CH28) [30] was used to analyze the distance between the N- and C-terminals and revealed that all four protein chains (A–D) had a very close distance between both termini at only 5.30–5.61 A° (Appendix A). Modification of the DPEase protein sequence via fusion with a 6xHis-tag and signal peptides, consisting of approximately 20–27 amino acids on the N- or C-terminals of the enzyme, may easily affect the structure of DPEase, resulting in a reduction in its solubility.

### 3.3. Construction of DPEase Localization Constructs

The expression vectors for DPEase localization were constructed using pET28a, a cytoplasm expression plasmid for the *E. coli* expression host. The expression vectors with N-terminal-selected signal peptides, OmpA and PelB (Figure 1), were fused to the synthesized DPEase gene under the control of the T7 promoter and then transformed into *E. coli* BL21(DE3), a T7 expression host. Kan^r^ colonies were picked and verified for the *DPEase* integration using PCR, as shown in Appendix A. The expected bands for pET28a + DPEase, pET28a + OmpA + DPEase, and pET28a + PelB + DPEase (1098, 1161, and 1166 bp, respectively) were detected, resulting in strains designated as Psicose, OmpA_Psicose, and PelB_Psicose, respectively. Representative strains were further confirmed for correct *DPEase* gene integration in the expression plasmid using sequencing before proceeding to the next experiment.

### 3.4. DPEase Localization Expression and Recombinant Enzymes from Various Compartments’ Activity Levels

The total protein concentration and DPEase enzyme activity of the 3 compartments (fermentation broth, periplasm fraction, and cell lysate) from each expression strain were analyzed. The highest total protein concentration was observed in the cell lysate in all samples, followed by the fermentation broth and periplasm fraction samples (Table 4). The fermentation broth of the OmpA_Psicose and PelB_Psicose strains exhibited protein concentrations 4.6-fold and 5.8-fold higher than Psicose, respectively. A similar result was observed in the periplasm fraction, with a 2.7-fold higher protein concentration in strains containing the OmpA signal peptide. These results indicated that OmpA signal peptides in the expression vector were able to enhance protein localization to the periplasmic space, while the PelB signal peptides promoted the extracellular secretion of proteins. However, the enzyme concentrations in the fermentation broth and periplasm fractions were too low to detect, as shown by the SDS-PAGE results (Figure 2A–C).

In this experiment, the activity of recombinant enzymes from various compartments was also checked to compare the DPEase activity of the three different plasmid constructs in each compartment. The DPEase activity was detected in all cell lysate samples with the activity ranging from 1.2 to 1.6 unit/mL (Table 4); however, the cell lysate sample from the Psicose strain exhibited the highest specific activity of 1.3 unit/mg. All non-induced samples showed no DPEase activity in all cell compartments. These results indicated the successful construction of DPEase expression strains using the three different plasmid constructs inside the cells. In contrast, no DPEase activity could be detected in the fractions from the fermentation broth and periplasm in all samples.

The SDS-PAGE analysis was performed to confirm the DPEase expression level in each compartment (Figure 2A–C). The Psicose strain had a high DPEase concentration in the cell lysate fraction with an intense protein band at 33 kDa (Figure 2A), which corresponded to the calculated DPEase size, whereas only faint bands were detected in the fermentation broth and periplasm fractions. Notably, in the cell debris of the Psicose and PelB_Psicose samples, an intense protein band was detected corresponding to DPEase, which could have been an insoluble expressed recombinant protein (Figure 2A,C). Thus, in this experiment, two strategies for the reduction in insoluble protein were investigated. The first strategy aimed to increase protein secretion and solubility using Triton X-100. Secondly, low-temperature expression was also studied to reduce the protein expression rate and mitigate inclusion body formation.

### 3.5. Effect of Triton X-100 on the Localization Expression of DPEase

Based on the results above, high protein concentrations were detected at the position of DPEase in cell debris from both the Psicose stain and the PelB_Psicose strain, reflecting a high amount of insoluble protein, such as inclusion bodies. To address this issue, Triton X-100, a non-ionic surfactant that can promote the extracellular enzyme without negatively affecting *E. coli* cell growth [31], was investigated. The supplement of Triton X-100 aimed to enhance the secretion of DPEase recombinant protein by minimizing protein accumulation in cell debris.

Cultivation of recombinant strains with 0.1% (*v*/*v*) Triton X-100 resulted in higher protein secretion in all samples, with a higher protein concentration in the fermentation broth (Table 5) compared to the cultured condition without Triton X-100 (Table 4). Notably, in the PelB_Psicose-X strain, the enzyme activity was detected in both the cell lysate and the fermentation broth.

The results were clearly evident in the SDS-PAGE analysis (Figure 2D–F). Cultivation of PelB_Psicose with Triton X-100 supplementation not only enhanced the secretion of proteins but also produced higher concentrations of DPEase in the fermentation broth and cell lysate, particularly in the PelB_Psicose strain (Figure 2F). This result confirmed the success of using PelB for the secretion of DPEase with Triton X-100 as a secretion enhancer.

### 3.6. Effect of Low Temperature on DPEase Expression

The expression of DPEase at low temperature was also studied to minimize the formation of inclusion bodies. The Psicose strain was cultured at various temperatures (20 °C, 25 °C, 30 °C, and 37 °C). The highest recombinant protein expression was detected at 20 °C cultivation (Figure 3A), while lower expression was observed at 25 and 30 °C. No DPEase was detected when cultured at 37 °C.

This confirmed that low-temperature expression was suitable for DPEase, which could increase enzyme production and enhance the solubility of the enzyme. Expression at low temperatures is a common strategy used to increase the solubility of heterologous protein expression in *E. coli* [32]. Based on this result, the recombinant DPEase strains were cultured at 20 °C and the protein expression in each compartment was compared.

The enzyme activity results revealed that cultivation at a low temperature (20 °C) with 0.1% (*v*/*v*) Triton X-100 enhanced DPEase solubility and reduced misfolded protein, which showed higher specific activity of the enzyme in cell lysate samples, with values of 0.76, 1.18, and 1.18 unit/mg in Psicose, OmpA_Psicose, and PelB_Psicose, respectively (Table 6). Notably, the PelB_Psicose strain exhibited a 3-fold higher level of DPEase secretion into the supernatant with a specific activity of 0.26 unit/mg. This suggested that the slow transcription and translation of DPEase with low-temperature expression enhanced properly folded enzymes and reduced protein aggregation, which was confirmed by the lower intensity of the DPEase protein band in the cell lysate and a more intense protein band in the SDS-PAGE at the position of DPEase in the fermentation broth sample (Figure 3B–D).

## 4. Discussion

The use of *E. coli* as an expression host is well established due to its ability to achieve high levels of target protein expression at relatively low production costs. Over the years, considerable efforts have been directed toward developing systems for the localization and secretion of recombinant proteins in *E. coli* [33]. These systems aim to enhance protein solubility, folding, and stability by leveraging the oxidative environment of the periplasmic space and extracellular milieu, as well as the availability of chaperones [34,35]. Despite these advancements, the successful localization of DPEase in *E. coli* has remained elusive and unreported. In recent years, machine learning algorithms have played a crucial role in predicting recombinant protein production [28,36,37,38], thereby elucidating the intrinsic constraints associated with individual proteins. In silico studies have revealed that the fusion of signal peptides can significantly influence the solubility of DPEase, with different signal peptides exhibiting varying effects. This observation can be attributed to the structural characteristics of DPEase, which demonstrate a proximity between the N- and C-terminal ends. Modification of these terminal ends through the addition of long signal peptides or tag sequences facilitates interactions between them [39]. Experimental evidence supports the pivotal role of protein termini in stability, function, catalytic activity, and solubility [40,41,42,43,44,45].

In this experiment, the gene encoding DPEase from *A. tumefaciens* C58 also performed codon optimization before the cloning step (Appendix A). The preliminary study revealed that the *E. coli* recombinant strain carrying the pFLAG-CTS expression vector with the native gene exhibited an intracellular DPEase activity level of 0.057 unit/mL, which was lower than that observed in the *A. tumefaciens* wild-type strain (0.134 unit/mL) (unpublished data). This problem has been reported in various studies on heterologous gene expression in several expression hosts, including *E. coli*, *Saccharomyces cerevisiae*, *Drosophila*, *Caenorhabditis elegans*, *Synechococcus*, and *Prochlorococcus* [46]. The expression of native genes could present codon bias of rare usage codons, which results in suppression of the protein expression level [47]. This suggests that the native gene from *A. tumefaciens* may use *E. coli* rare usage codons, which could reduce the efficiency of translation or even disengage the translational machinery [48]. After codon optimization, the codon usage bias in *E. coli* was addressed by improving the codon adaptation index from 0.63 to 0.96, confirming the existence of the rare usage codon in this native gene. The success of this experiment with a codon-optimized DPEase gene from *A. tumefaciens* demonstrated the importance of codon optimization for enhancing heterologous gene expression in *E. coli*.

When comparing widely used signal peptides for *E. coli*, such as OmpA [49], with the best predicted PelB for constructing expression plasmids for DPEase localization, intriguing differences emerged. The recombinant strain with PelB exhibited enzyme activity in the supernatant but not in the periplasm fraction. This phenomenon is observed in certain cases involving type II signal peptides which export the target protein to the periplasmic space via the Sec pathway, where protein folding occurs [49,50]. In these instances, extracellular secretion relies on the common Sec system, encompassing both the characteristics of the PelB signal peptide and the target protein itself [49]. Moreover, the presence of the target enzyme in the supernatant may be attributed to its synthesis in the cytoplasm and subsequent export to the periplasmic space [49,51], potentially increasing osmotic pressure within the *E. coli* cell and leading to outer membrane rupture, releasing proteins into the supernatant [52]. Conversely, the recombinant strain carrying the OmpA signal peptide did not exhibit detectable enzyme activity in either the fermentation broth or the periplasmic fractions. SDS-PAGE analysis of protein expression in the OmpA_Psicose strain revealed a prominent protein band larger than the size of DPEase (Figure 2B,E), which was absent in the Psicose and PelB_Psicose strain protein expressions. The increased molecular weight of this protein band suggested a fusion of DPEase with the OmpA signal peptide. Insufficient cleavage of the OmpA signal peptide from *Staphylococcus aureus* in the *E. coli* expression host could explain this, as reported with certain signal peptides [53,54]. The formation of this protein fusion with OmpA could lead to misfolding of DPEase, affecting its biochemical properties and inhibiting its enzyme activity [20,55].

The expression of DPEase in *E. coli* revealed the formation of inclusion bodies, especially at 37 °C cultivation. Recombinant protein expression in *E. coli* often leads to the formation of insoluble protein aggregates, which are known as inclusion bodies, consisting of misfolded and non-functional proteins [46,55,56,57]. The high expression rate driven by the T7 promoter, coupled with the absence of chaperones, results in insufficient protein folding and contributes to the production of misfolded proteins [56,57]. In addition, a high protein expression rate leads to increased protein concentration in the cytoplasm or periplasm, which reduces the solubility of the protein and promotes protein aggregation [58,59]. To overcome this issue, lowering the culturing temperature slows down cell growth, thus leading to a reduced rate of transcription and translation and minimizing the amounts of misfolded proteins and protein aggregation [47,56,59]. Moreover, supplementation with 0.1% (*v*/*v*) Triton X-100 in the culture medium could increase secretion efficiency. The higher extracellular DPEase level was detected due to the efficient impairment of *E. coli’s* outer membrane, enhancing cell permeability [31]. A similar result was reported in pullulanase expression with 0.5% (*v*/*v*) Triton X-100, which solubilized the insoluble fraction and increased extracellular enzyme activity [60].

Although the safety evaluation results of D-psicose synthesized from recombinant DPEase produced by *E. coli* BL21(DE3) confirmed the safety of our recombinant DPEase (unpublished data), endotoxin contamination remains a concern when *E. coli* is used as the expression host. However, endotoxin contamination may still be a concern when *E. coli* is used as the expression host. The secretion of DPEase not only reduces downstream processing costs but also decreases the risk of endotoxin contamination by preventing the release of LPS endotoxins from the cell membrane during cell disruption [61]. Furthermore, concentrating the supernatant with a 10 kDa cut-off membrane effectively removes LPS contamination, as LPS typically has a size of about 10 kDa, and membranes with a 6–10 kDa molecular weight cut-off (MWCO) are used for endotoxin removal [62,63]. This approach ensures minimal contamination and effective LPS removal from the DPEase.

## 5. Conclusions

In conclusion, this study demonstrates the successful intracellular and extracellular expression of DPEase in *E. coli*, facilitated by the utilization of in silico analysis to evaluate signal peptides and protein structure. Specifically, the engineered *E. coli* recombinant strain, incorporating the PelB signal peptide, exhibited notably higher extracellular DPEase activities compared to the wild type. Additionally, employing low-temperature expression conditions enhanced enzyme solubility and minimized inclusion body formation. The fusion of a signal peptide to DPEase proved crucial for efficient enzyme secretion, with the selection of an appropriate signal peptide validated by computational predictions and the experimental results. These findings not only optimize downstream enzyme processes but also hold promise for advancing whole-cell immobilization techniques reliant on efficient protein secretion mechanisms. This advancement has the potential to expand the practical applications of DPEase and other target proteins. Moreover, given recent concerns regarding the potential virulence of certain signal peptides, such as OmpA in pathogenic *Enterobacteriaceae*, the exploration of alternative signal peptides becomes imperative for ensuring the safety and efficacy of engineered microbial systems in biotechnological applications. This study underscores the importance of this ongoing research endeavor, with promising prospects emerging for the development of novel signal peptide strategies in enzyme secretion.

## Figures and Tables

**Figure 1 microorganisms-12-01574-f001:**
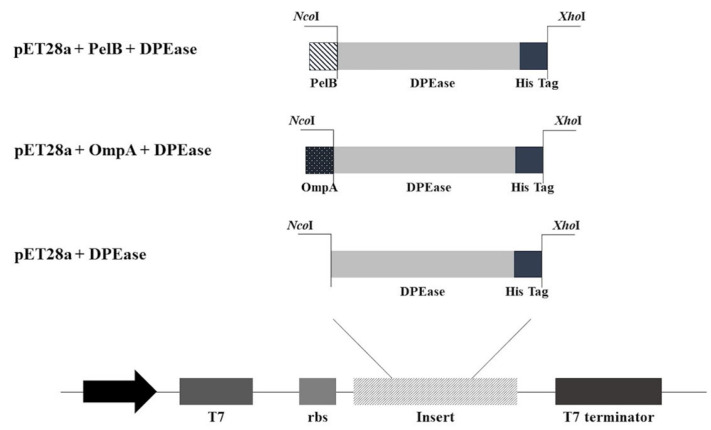
Plasmid construction map for this study. The synthesized *DPEase* genes were cloned into pET28a with and without OmpA and PelB signal peptides at the 5′ *Nco*I and 3′ *Xho*I sites. The expression of all constructs was under the control of the T7 promoter.

**Figure 2 microorganisms-12-01574-f002:**
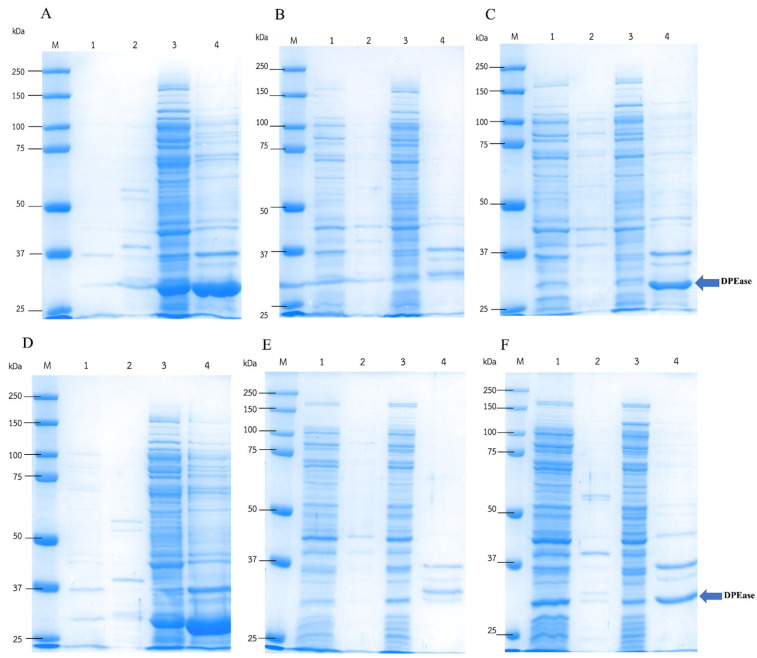
SDS-PAGE analysis of recombinant DPEase in the *E. coli* Psicose strain (**A**,**D**), the OmpA_Psicose strain (**B**,**E**), and the PelB_Psicose strain (**C**,**F**) cultured in LB medium without (**A**–**C**) and with (**D**–**F**) 0.1% (*v*/*v*) Triton X-100. The samples from various compartments were analyzed. (Lane M: protein marker; lane 1: fermentation broth fraction; lane 2: periplasm fraction; lane 3: cell lysate fraction and lane 4: cell debris).

**Figure 3 microorganisms-12-01574-f003:**
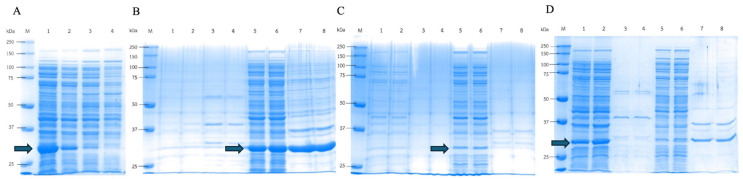
Effect of temperature on DPEase expression. (**A**) SDS-PAGE analysis of recombinant DPEase in the *E. coli* Psicose strain cultured at 20 °C (lane 1), 25 °C (lane 2), 30 °C (lane 3), and 37 °C (lane 4). SDS-PAGE analysis of DPEase expression at 20 °C of the Psicose strain (**B**), the OmpA_Psicose strain (**C**), and the PelB_Psicose strain (**D**) with 0.1% (*v*/*v*) Triton X-100. The samples from various compartments were analyzed. (lane M: protein marker; lane 1, 2: fermentation broth fraction; lane 3, 4: periplasm fraction; lane 5, 6: cell lysate fraction and lane 7, 8: cell debris). Blue arrows represent the DPEase position.

**Table 1 microorganisms-12-01574-t001:** Bacterial strains and plasmids used in this experiment.

Strain and Plasmid	Relevant Features	Source
Strain		
*Escherichia coli* BL21(DE3)	Expression host	Lab stock
Neb5α	Cloning host	NEB
Psicose	*E. coli* BL21 harboring pET28a + DPEase	This work
OmpA_Psicose	*E. coli* BL21 harboring pET28a + OmpA + DPEase	This work
PelB_Psicose	*E. coli* BL21 harboring pET28a + PelB + DPEase	This work
Plasmid		
pET28a	Kan^r^; T7 *lac*	Novagen
pET28a + DPEase	Kan^r^; pET28a derivative; D-psicose-3-epimerase biosynthesis gene	This work
pET28a + OmpA + DPEase	Kan^r^; pET28a + OmpA derivative; D-psicose-3-epimerase biosynthesis gene	This work
pET28a + PelB + DPEase	Kan^r^; pET28a + PelB derivative; D-psicose-3-epimerase biosynthesis gene	This work

**Table 2 microorganisms-12-01574-t002:** Cleavability and secretion pathway in the *E. coli* prediction of DPEase fused with signal peptides using SignalP 6.0.

Gene Construct	Secretion Pathway				Amino Acid Cleavage Position	Probability
	Other	Sec/SPI	Sec/SPII	Tat/SPI		
DPEase	1.000000	0.000000	0.000000	0.000000	nd	nd
draA_DPEase	0.000249	0.999048	0.000181	0.000173	21–22	0.9779
faeG_DPEase	0.000277	0.999000	0.000183	0.000177	21–22	0.9727
fedA_DPEase	0.000268	0.999026	0.000198	0.000165	21–22	0.9714
flgI_DPEase	0.000194	0.999163	0.000146	0.000153	20–21	0.9767
malE_DPEase	0.000265	0.999029	0.000175	0.000178	26–27	0.9735
OmpA_DPEase	0.000235	0.999069	0.000182	0.000177	21–22	0.9777
pbpG_DPEase	0.00021	0.99914	0.000162	0.000167	25–26	0.9754
PelB_DPEase	0.000312	0.99895	0.000185	0.000180	22–23	0.9785
xylF_DPEase	0.000245	0.999111	0.00017	0.000158	23–24	0.9762
yncJ_DPEase	0.000255	0.999044	0.000186	0.000168	22–23	0.9759
zraP_DPEase	0.000234	0.999058	0.000168	0.000178	26–27	0.9737
ampC_DPEase	0.000729	0.617677	0.380953	0.000254	19–20	0.5919
OmpC_DPEase	0.00022	0.999148	0.000161	0.000160	21–22	0.9732
STII_DPEase	0.000259	0.99905	0.000171	0.000166	23–24	0.9771
Ompf_DPEase	0.000262	0.999009	0.000171	0.000186	22–23	0.9786
lamB_DPEase	0.000262	0.999078	0.000165	0.000188	25–26	0.9694
araF_DPEase	0.000268	0.999022	0.000180	0.000171	23–24	0.9767
nmpc_DPEase	0.000248	0.999052	0.000167	0.000173	23–24	0.9788
ppiA_DPEase	0.000221	0.999077	0.000163	0.000178	24–25	0.9728
yaaI_DPEase	0.000481	0.954444	0.044384	0.000234	23–24	0.9212
glnH_DPEase	0.00024	0.999078	0.000160	0.000169	22–23	0.9795
rna_DPEase	0.00029	0.999048	0.000184	0.000160	23–24	0.9723
DsbC_DPEase	0.000237	0.999039	0.000214	0.000164	20–21	0.9773
rbsB_DPEase	0.000212	0.999116	0.000161	0.000185	25–26	0.9008
gfcA_DPEase	0.000299	0.998936	0.000200	0.000179	21–22	0.9765

nd: not detected. The shaded yellow-green colors represent the likelihood of gene constructs in each secretion pathway and the probability of cleavage positions.

**Table 3 microorganisms-12-01574-t003:** Localization and solubility prediction in *E. coli* of DPEase fused with signal peptides using DeepLogPro 1.0 and the Protein-Sol server.

Gene Construct	Cell Wall & Surface	Extra-Cellular	Intra-Cellular	Cytoplasmic Membrane	Outer Membrane	Periplasm	Solubility
DPEase	0	0.0002	0.9936	0.0002	0.0053	0.0008	0.919
draA_DPEase	0	0.0006	0.0232	0.0002	0.0458	0.9302	0.834
faeG_DPEase	0	0.0035	0.0088	0.0001	0.0248	0.9627	0.823
fedA_DPEase	0	0.0019	0.0286	0.0004	0.0516	0.9175	0.802
flgI_DPEase	0	0.0011	0.0374	0.0007	0.0319	0.9289	0.829
malE_DPEase	0	0.0010	0.0015	0.0001	0.0072	0.9903	0.798
OmpA_DPEase	0	0.0026	0.0115	0.0002	0.0283	0.9575	0.797
pbpG_DPEase	0	0.0021	0.0306	0.0005	0.0272	0.9396	0.799
PelB_DPEase	0	0.0015	0.0143	0.0002	0.0106	0.9919	0.890
xylF_DPEase	0	0.0016	0.0055	0.0002	0.0163	0.9764	0.867
yncJ_DPEase	0	0.0020	0.0157	0.0004	0.0261	0.9557	0.809
zraP_DPEase	0	0.0015	0.0008	0.0001	0.0057	0.9733	0.807
ampC_DPEase	0.0001	0.0053	0.0185	0.0003	0.0533	0.9225	0.835
OmpC_DPEase	0	0.0022	0.0149	0.0003	0.0339	0.9486	0.842
STII_DPEase	0.0001	0.0018	0.0310	0.0015	0.0820	0.8836	0.816
Ompf_DPEase	0	0.0013	0.0348	0.0002	0.0269	0.9368	0.818
lamB_DPEase	0	0.0025	0.0143	0.0002	0.0240	0.9589	0.832
araF_DPEase	0	0.0006	0.0064	0.0001	0.0185	0.9745	0.802
nmpc_DPEase	0	0.0007	0.0035	0.0001	0.0154	0.9803	0.800
ppiA_DPEase	0	0.0003	0.0041	0.0001	0.007	0.9885	0.767
yaaI_DPEase	0	0.0016	0.0256	0.0004	0.0163	0.9562	0.822
glnH_DPEase	0	0.0012	0.0044	0.0002	0.0287	0.9656	0.868
rna_DPEase	0	0.0015	0.0414	0.0004	0.0405	0.9162	0.864
DsbC_DPEase	0.0001	0.0016	0.0725	0.0006	0.0824	0.8429	0.850
rbsB_DPEase	0	0.0012	0.0065	0.0002	0.0261	0.9659	0.827
gfcA_DPEase	0	0.0007	0.0200	0.0002	0.0164	0.9627	0.800

The shaded yellow-green colors indicate the localization probability and predicted solubility scale of each gene constructs.

**Table 4 microorganisms-12-01574-t004:** Protein concentration and enzyme activity of recombinant enzymes from various compartments.

Sample	Fraction	Protein Conc. (mg/mL)	DPEase Activity (unit/mL)	Specific Activity (unit/mg)
Psicose	Fermentation broth	0.297 ± 0.004	ND	ND
	Cell lysate	2.572 ± 0.114	1.608 ± 0.043	0.625
	Periplasm fraction	0.085 ± 0.002	ND	ND
OmpA_Psicose	Fermentation broth	1.390 ± 0.043	ND	ND
	Cell lysate	1.371 ± 0.051	1.243 ± 0.138	0.907
	Periplasm fraction	0.231 ± 0.006	ND	ND
PelB_Psicose	Fermentation broth	1.709 ± 0.031	ND	ND
	Cell lysate	0.835 ± 0.015	1.379 ± 0.034	1.651
	Periplasm fraction	0.041 ± 0.005	ND	ND

ND: not detected.

**Table 5 microorganisms-12-01574-t005:** Effect of Triton X-100 supplementation on protein secretion and enzyme activity.

Sample	Fraction	Protein Conc. (mg/mL)	DPEase Activity (unit/mL)	Specific Activity (unit/mg)
Psicose-X	Fermentation broth	1.341 ± 0.054	ND	ND
	Cell lysate	2.531 ± 0.068	1.019 ± 0.048	0.403
	Periplasm fraction	0.069 ± 0.002	ND	ND
OmpA_Psicose-X	Fermentation broth	1.709 ± 0.031	ND	ND
	Cell lysate	1.835 ± 0.015	1.609 ± 0.073	0.877
	Periplasm fraction	0.041 ± 0.005	ND	ND
PelB_Psicose-X	Fermentation broth	1.999 ± 0.030	0.163 ± 0.047	0.082
	Cell lysate	1.438 ± 0.184	1.545 ± 0.085	1.074
	Periplasm fraction	0.062 ± 0.024	ND	ND

ND: not detected.

**Table 6 microorganisms-12-01574-t006:** Effect of low induction temperature on DPEase secretion and enzyme activity.

Sample	Fraction	Protein Conc. (mg/mL)	DPEase Activity (unit/mL)	Specific Activity (unit/mg)
Psicose-X	Fermentation broth	0.748 ± 0.028	ND	ND
	Cell lysate	2.490 ± 0.132	1.897 ± 0.053	0.75
	Periplasm fraction	0.074 ± 0.009	ND	ND
OmpA_Psicose-X	Fermentation broth	0.923 ± 0.060	ND	ND
	Cell lysate	1.109.0 ± 0.058	1.309 ± 0.031	1.18
	Periplasm fraction	0.02 ± 0.001	ND	ND
PelB_Psicose-X	Fermentation broth	1.936 ± 0.017	0.512 ± 0.047	0.26
	Cell lysate	1.157 ± 0.038	1.362 ± 0.078	1.18
	Periplasm fraction	0.030 ± 0.008	ND	ND

ND: not detected.

## Data Availability

Data are contained within the article and Appendix A.

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
