# Peer review of "In Silico Analysis and Development of the Secretory Expression of D-Psicose-3-Epimerase in Escherichia coli"

_microorganisms, 2024, doi:10.3390/microorganisms12081574_

Round 1
Reviewer 1 Report
Comments and Suggestions for Authors
The results of the extracellular expression of D-psicose-3-epimerase, an enzyme of industrial importance, are very valuable.
However, since synthesis of vectors has recently become very convenient and it is easy to make vectors, culture them, and check SDS-PAGE of the supernatant, more expression experiments using signal peptides other than OmpA and PelB should be performed to confirm protein production in the supernatant. Otherwise, we will not know if PelB was really suitable or if this signal peptide selection method is a good method.
Synthesis of vectors has become so conven
Other comments:
1. In the text, Figures 2 and 3 are misrepresented, which makes them difficult to understand. There is an explanation for Figure 4, but there is no Figure 4.
2. 3.2 The DPEase structure analysis, L241-244: The meaning of the sentence is unclear.
Author Response
|
Response to Reviewer
|
||
|
1. Summary |
|
|
|
Thank you very much for taking the time to review this manuscript. Please find the detailed responses below and the corresponding revisions highlighted in the re-submitted files.
|
||
|
2. Point-by-point response to Comments and Suggestions for Authors |
||
|
|
||
|
Comments 1: However, since synthesis of vectors has recently become very convenient and it is easy to make vectors, culture them, and check SDS-PAGE of the supernatant, more expression experiments using signal peptides other than OmpA and PelB should be performed to confirm protein production in the supernatant. Otherwise, we will not know if PelB was really suitable or if this signal peptide selection method is a good method.
Response 1: Thank you for your suggestion! The aim of this work is to demonstrate the feasibility of using in-silico analysis for selecting signal peptides for E. coli. This is a critical step for E. coli secretion expression, particularly for DPEase. We chose the best score from the prediction (PelB) and compared it to the commonly used signal peptide for E. coli (OmpA), which has recently been reported as an endotoxin-related gene. Due to budget constraints, we could not synthesize vectors with all selected signal peptides. However, the findings from this manuscript will serve as a foundation for further research using in-silico analysis for other enzymes. This will help to develop and validate the significance of prediction models in signal peptide selection.
|
||
|
Comments 2: 1. In the text, Figures 2 and 3 are misrepresented, which makes them difficult to understand. There is an explanation for Figure 4, but there is no Figure 4. Response 2: The Figure 2, 3 and 4 in text has been carefully revised and rewritten as suggested (line 290, 307, 312, 365, 395, 495)
|
||
|
Comments 3: 2. 3.2 The DPEase structure analysis, L241-244: The meaning of the sentence is unclear. Response 3: The sentence describe about DPEase structure has been rewritten as “Modification of the DPEase protein sequence by fusion with a 6xHis-tag and signal peptides, consisting of approximately 20-27 amino acids on N- or C-terminals of the enzyme, may easily affect the structure of DPEase which resulting in a reduction of its solubility.” for more clarity as suggested. (line 243-246) |
||

Reviewer 2 Report
Comments and Suggestions for Authors
An interesting paper. The authors can consider including more detailed experimental controls and validation steps to strengthen their findings. Specifically, they can incorporate negative controls to account for background activity and ensure the specificity of DPEase expression. Additionally, the study would benefit from a more thorough discussion of potential off-target effects of signal peptides and the impact of various expression conditions on cell viability. The authors should also explore the use of alternative host organisms or co-expression systems to enhance protein yield and stability. Including a comparative analysis of different signal peptides' effects on secretion efficiency and protein folding could provide deeper insights. Furthermore, more extensive characterization of the recombinant protein, including its structural and functional stability under different environmental conditions, would be valuable. Lastly, the inclusion of a cost-benefit analysis of the proposed secretion strategies in an industrial setting could make the study more applicable for biotechnological applications.
In line 2, "In-Silico" should be in italic
In line 36: how are stimulation of reactive oxygen species (ROS) and antioxidant behavior related?
In line 42, "very small amounts": please quantify
Smaller comments:
In line 12, "intracellularly expression" should be corrected to "intracellular expression".
In line 14, "Secretion expression" should be corrected to "Secretory expression".
In line 17, the word "through" is missing an "r".
In line 18, "PelB as the most effective for DPEase localization" should be "PelB as the most effective signal peptide for DPEase localization".
In line 20, "mitigated inclusion body formation" should be "mitigated the formation of inclusion bodies".
In line 21, "enhancing DPEase solubility" should be "thus enhancing DPEase solubility".
In line 25, "secretion strategies promise further enhancements" should be "secretion strategies promise further enhancement".
In the Keywords section, "Psicose," should be "Psicose;".
In line 33, "D-xylose, D-ribose, and L- arabinose" should be "D-xylose, D-ribose, and L-arabinose".
In line 34, "beneficial characteristics, such as antioxidant activity, anti-inflammatory activity, cancer and tumor inhibition, stimulation of reactive oxygen species (ROS)," should be revised for clarity and consistency.
In line 45, "GRAS" should be spelled out as "Generally Recognized As Safe (GRAS)" on first use.
In line 48, "reduce visceral fat mass" should be "reduced visceral fat mass".
In line 49, "hypoglycemic properties and therapeutic effects" should be "hypoglycemic properties and therapeutic effects".
In line 52, "conversion of D-fructose to D-psicose" should be "the conversion of D-fructose to D-psicose".
In line 60, "with some limitation" should be "with some limitations".
In line 63, "secretion systems [18], has not been" should be "secretion systems [18], has not yet been".
In line 66, "Although OmpA has reported" should be "Although OmpA has been reported".
In line 67, "Consequently, the exploration of alternative signal peptides for DPEase secretion has become imperative, with promising prospects emerging." should be revised for clarity.
In the Introduction section, improve clarity and flow between paragraphs.
In line 80, "E. coli Neb5a was used as an intermediate cloning host and along with the E. coli BL21(DE3)" should be "E. coli Neb5a was used as an intermediate cloning host, and along with the E. coli BL21(DE3)".
The list in Table 1 should be checked for formatting consistency.
In line 107, "New Jersey, USA" should be "New Jersey, USA,".
In line 109, "standard digestion and ligation cloning procedures" should be "standard digestion and ligation procedures".
In line 117, "localization expression of DPEase" should be "localization and expression of DPEase".
In line 120, "containing antibiotic" should be "containing the appropriate antibiotic".
In line 122, "antibiotic and cultivated at 37°C" should be "antibiotic, and cultivated at 37°C".
In line 124, "concentration of 1 mM and further" should be "concentration of 1 mM, and further".
In line 125, "at 30°C overnight (16 h)" should be "at 30°C for 16 hours".
In line 127, "prepared into 3 fractions: fermentation broth, periplasm fraction, and cell lysate" should be "prepared into three fractions: fermentation broth, periplasm fraction, and cell lysate".
In line 128, "sample: The cell-free supernatants" should be "sample: the cell-free supernatants".
In line 135, "4°C. The extracted" should be "4°C, the extracted".
In line 140, "pH 7.5. The cells" should be "pH 7.5, the cells".
In line 141, "for 2 min) on ice" should be "for 2 minutes on ice".
In line 143, "30 min, 4°C" should be "30 minutes at 4°C".
In line 145, "albumin as a standard protein and the size" should be "albumin as a standard, and the size".
In line 151, "localization expression of DPEase" should be "localization and expression of DPEase".
In line 157, "were measured" should be "was measured".
In line 160, "on DPEase expression" should be "of DPEase expression".
In line 162, "various temperature" should be "various temperatures".
In line 165, "30°C for 16 h" should be "30°C for 16 hours".
In line 165, "at 4 different temperatures" should be "at four different temperatures".
In line 166, "were analyzed" should be "was analyzed".
In line 171, "measured follow the protocol" should be "measured following the protocol".
In line 174, "incubation at 4°C with 1 mM Mn2+ in 100 mM Tris-HCl" should be "incubation at 4°C with 1 mM Mn2+ in 100 mM Tris-HCl buffer".
In line 175, "for 10 min and stopped" should be "for 10 minutes and stopped".
In line 176, "10 min. The D-psicose" should be "10 minutes. The D-psicose".
In line 180, "Twenty microliters of each sample were injected and the elution was performed" should be "Twenty microliters of each sample was injected, and the elution was performed".
In line 182, "50°C and pH 8.0 [12]" should be "50°C and pH 8.0".
In line 183, "specific activity of DPEase (unit/mg) was calculated from DPEase activity (unit/ml) divided by concentration of protein (mg/mL)" should be "specific activity of DPEase (unit/mg) was calculated by dividing DPEase activity (unit/mL) by protein concentration (mg/mL)".
In line 189, "greater than 85%) in E. coli from previous report" should be "greater than 85%) in E. coli from a previous report".
In line 191, "Table 2. Most of the selected signal" should be "Table 2. Most selected signal".
In line 198, "Table 2 Cleavability" should be "Table 2: Cleavability".
In line 207, "Table 3 Localization" should be "Table 3: Localization".
In line 223, "reduction in DPEase solubility" should be "reduction in the solubility of DPEase".
In line 223, "solubility with a value of 0.89" should be "solubility, with a value of 0.89".
In line 228, "secretion efficiency for in vitro analysis, validating" should be "secretion efficiency for in vitro analysis, thereby validating".
In line 234, "DPEase. So, the DPEase" should be "DPEase; therefore, the DPEase".
In line 239, "the all four protein" should be "all four protein".
In line 244, "DPEase structure analysis" should be "DPEase structure analysis.".
In line 249, "BL21(DE3), T7 expression host" should be "BL21(DE3), a T7 expression host".
In line 251, "PCR, as shown in the Figure S2" should be "PCR, as shown in Figure S2".
In line 254, "with N- terminal selected" should be "with N-terminal selected".
In line 273, "under control of T7 promoter" should be "under the control of the T7 promoter".
In line 276, "from various compartment activity levels" should be "from various compartments' activity levels".
In line 278, "samples. The highest" should be "samples, the highest".
In line 282, "Psicose, respectively. A similar result" should be "Psicose, respectively; a similar result".
In line 285, "samples. However, the enzyme" should be "samples; however, the enzyme".
In line 287, "fractions were too low to be detected" should be "fractions were too low to detect".
Author Response
|
Response to Reviewer
|
||
|
1. Summary |
|
|
|
Thank you very much for taking the time to review this manuscript. Please find the detailed responses below and the corresponding revisions highlighted in the re-submitted files.
|
||
|
2. Point-by-point response to Comments and Suggestions for Authors |
||
|
|
||
|
Comments 1: The authors can consider including more detailed experimental controls and validation steps to strengthen their findings. Specifically, they can incorporate negative controls to account for background activity and ensure the specificity of DPEase expression. Response 1: The DPEase activity measurement has been updated as suggested. The DPEase activity was performed by checked the conversion efficiency of D-fructose to D-psicose, which is very specific reaction occurs only by DPEase activity. The HPLC was used to detected D-fructose and D-Psicose in the tested reaction comparing to standard. Due to it’s specificity, no background reaction detected if no DPEase present in the system. So, the DPEase activity measurementmethod has been updated for more understanding as suggested. (line 150-152).
|
||
|
Comments 2: Additionally, the study would benefit from a more thorough discussion of potential off-target effects of signal peptides and the impact of various expression conditions on cell viability. Response 2: The potential off-target effects of signal peptides has been discussed in the discussion part (line 447-455), and the impact of various expression conditions on cell viability is going to report in the next manuscript.
|
||
|
Comments 3: The authors should also explore the use of alternative host organisms or co-expression systems to enhance protein yield and stability. Response 3: Thank you for the suggestion, we have also been studying the expression of DPEase in yeast expression system.
|
||
|
Comments 4: Furthermore, more extensive characterization of the recombinant protein, including its structural and functional stability under different environmental conditions, would be valuable. Response 4:Thank you for your suggestion. The characterization of DPEase enzyme has been characterized for the best condition. This will be reported in the next experiment along with safety evaluation of D-psicose.
|
||
|
Comments 5: Lastly, the inclusion of a cost-benefit analysis of the proposed secretion strategies in an industrial setting could make the study more applicable for biotechnological applications. Response 5: This information has been discussed in the discussion part (line 503-505).
|
||
|
Comments 6: Lastly, the inclusion of a cost-benefit analysis of the proposed secretion strategies in an industrial setting could make the study more applicable for biotechnological applications. Response 6: This information has been discussed in the discussion part (line 503-505).
|
||
|
In line 2, "In-Silico" should be in italic Response : The word In-silico has been corrected in Italic (line 2).
|
||
|
In line 36: how are stimulation of reactive oxygen species (ROS) and antioxidant behavior related? Response : The sentence has been corrected and rewritten as “reduction of reactive oxygen species” (line 36).
|
||
|
In line 42, “very small amounts”: please quantify Response : The amount of D-psicose in nature sources has been quantified as “about 0.38 and 0.29 mg/g of raisin and dried fig” (line 42-43).
|
||
|
In line 12, "intracellularly expression" should be corrected to "intracellular expression". Response : The word has been corrected as suggested (line 12).
|
||
|
In line 14, "Secretion expression" should be corrected to "Secretory expression". Response : The word has been corrected as suggested (line 14).
|
||
|
In line 17, the word "through" is missing an "r". Response : The word has been corrected (line 17).
|
||
|
In line 18, "PelB as the most effective for DPEase localization" should be "PelB as the most effective signal peptide for DPEase localization". Response : The word “signal peptide” has been added as suggested (line 18).
|
||
|
In line 20, "mitigated inclusion body formation" should be "mitigated the formation of inclusion bodies". Response : The sentence has been revised as suggested (line 21-22).
|
||
|
In line 21, "enhancing DPEase solubility" should be "thus enhancing DPEase solubility". Response : The word “thus” has been added as suggested (line 21).
|
||
|
In line 25, "secretion strategies promise further enhancements" should be "secretion strategies promise further enhancement". Response : The word has been corrected (line 25).
|
||
|
In the Keywords section, "Psicose," should be "Psicose;". Response : The “;” has been corrected (line 27).
|
||
|
In line 33, "D-xylose, D-ribose, and L- arabinose" should be "D-xylose, D-ribose, and L-arabinose". Response : The sentence has been revised as suggested (line 32-33).
|
||
|
In line 34, "beneficial characteristics, such as antioxidant activity, anti-inflammatory activity, cancer and tumor inhibition, stimulation of reactive oxygen species (ROS)," should be revised for clarity and consistency. Response : The sentence has been corrected and rewrite as “reduction of reactive oxygen species” (line 36).
|
||
|
In line 45, "GRAS" should be spelled out as "Generally Recognized As Safe (GRAS)" on first use. Response : The sentence has been revised as suggested (line 45-46).
|
||
|
In line 48, "reduce visceral fat mass" should be "reduced visceral fat mass". Response : The word has been corrected (line 49).
|
||
|
In line 49, "hypoglycemic properties and therapeutic effects" should be "hypoglycemic properties and therapeutic effects". Response : The sentence has been revised as suggested (line 49-50).
|
||
|
In line 52, "conversion of D-fructose to D-psicose" should be "the conversion of D-fructose to D-psicose". Response : The sentence has been revised as suggested (line 52).
|
||
|
In line 60, "with some limitation" should be "with some limitations". Response : The word has been corrected (line 61).
|
||
|
In line 63, "secretion systems [18], has not been" should be "secretion systems [18], has not yet been". Response : The sentence has been revised as suggested (line 64).
|
||
|
In line 66, "Although OmpA has reported" should be "Although OmpA has been reported". Response : The sentence has been revised as suggested (line 65).
|
||
|
In line 67, "Consequently, the exploration of alternative signal peptides for DPEase secretion has become imperative, with promising prospects emerging." should be revised for clarity. Response : The sentence has been rewritten for more clarity as suggested “Consequently, the exploration of alternative signal peptides for DPEase secretion in E. coli could reduce the endotoxin contamination concerns from E. coli expression system and support the use of recombinant DPEase for industrial applications “(line 67-70). |
||
|
In the Introduction section, improve clarity and flow between paragraphs. Response : The introduction has been rewritten as suggested (line 36, 49-50, 67-70)
|
||
|
In line 80, "E. coli Neb5a was used as an intermediate cloning host and along with the E. coli BL21(DE3)" should be "E. coli Neb5a was used as an intermediate cloning host, and along with the E. coli BL21(DE3)". Response : The sentence has been revised as suggested (line 83).
|
||
|
The list in Table 1 should be checked for formatting consistency. Response : The list in the Table 1 has been checked as suggested.
|
||
|
In line 107, "New Jersey, USA" should be "New Jersey, USA,". Response : The word has not been corrected as suggested for the consistency throughout the manuscript.
|
||
|
In line 109, "standard digestion and ligation cloning procedures" should be "standard digestion and ligation procedures". Response : The sentence has been revised as suggested (line 111).
|
||
|
In line 117, "localization expression of DPEase" should be "localization and expression of DPEase". Response : The sentence has been revised as suggested (line 118).
|
||
|
In line 120, "containing antibiotic" should be "containing the appropriate antibiotic". Response : The sentence has been the same as the suggestion (line 122).
|
||
|
In line 122, "antibiotic and cultivated at 37°C" should be "antibiotic, and cultivated at 37°C". Response : The sentence has been revised as suggested (line 123).
|
||
|
In line 124, "concentration of 1 mM and further" should be "concentration of 1 mM, and further". Response : The sentence has been revised as suggested (line 126).
|
||
|
In line 125, "at 30°C overnight (16 h)" should be "at 30°C for 16 hours". Response : The word has not been corrected as suggested for the consistency throughout the manuscript (line 127).
|
||
|
In line 127, "prepared into 3 fractions: fermentation broth, periplasm fraction, and cell lysate" should be "prepared into three fractions: fermentation broth, periplasm fraction, and cell lysate". Response : The sentence has been the same as the suggestion (line 128-129).
|
||
|
In line 128, "sample: The cell-free supernatants" should be "sample: the cell-free supernatants". Response : The sentence has been revised as suggested (line 130).
|
||
|
In line 135, "4°C. The extracted" should be "4°C, the extracted". Response : The sentence has not been corrected as suggested due to more clarity with the separated sentence (line 138).
|
||
|
In line 140, "pH 7.5. The cells" should be "pH 7.5, the cells". Response : The sentence has not been corrected as suggested due to more clarity with the separated sentence (line 141).
|
||
|
In line 141, "for 2 min) on ice" should be "for 2 minutes on ice". Response : The “)” has been deleted as suggested (line 143).
|
||
|
In line 143, "30 min, 4°C" should be "30 minutes at 4°C". Response : The word has not been corrected as suggested for the consistency throughout the manuscript (line 144).
|
||
|
In line 145, "albumin as a standard protein and the size" should be "albumin as a standard, and the size". Response : The sentence has been revised as suggested (line 147).
|
||
|
In line 151, "localization expression of DPEase" should be "localization and expression of DPEase". Response : The sentence has been revised as suggested (line 154).
|
||
|
In line 157, "were measured" should be "was measured". Response : The word has been corrected (line 160).
|
||
|
In line 160, "on DPEase expression" should be "of DPEase expression". Response : The sentence has not been corrected as suggested due more clarity (line 162).
|
||
|
In line 162, "various temperature" should be "various temperatures". Response : The word has been corrected (line 164).
|
||
|
In line 165, "30°C for 16 h" should be "30°C for 16 hours". Response : The word has not been corrected as suggested for the consistency throughout the manuscript (line 168).
|
||
|
In line 165, "at 4 different temperatures" should be "at four different temperatures". Response : The word “four” has been revised as suggested (line 167).
|
||
|
In line 166, "were analyzed" should be "was analyzed". Response : The sentence has not been corrected as suggested due to more than sample were analyzed for this experiment (line 171).
|
||
|
In line 171, "measured follow the protocol" should be "measured following the protocol". Response : The sentence has been revised as suggested (line 174).
|
||
|
In line 174, "incubation at 4°C with 1 mM Mn2+ in 100 mM Tris-HCl" should be "incubation at 4°C with 1 mM Mn2+ in 100 mM Tris-HCl buffer". Response : The word “buffer” has been radded as suggested (line 176).
|
||
|
In line 175, "for 10 min and stopped" should be "for 10 minutes and stopped". Response : The word “min” has not been changed as suggested for the consistency throughout the manuscript (line 178).
|
||
|
In line 176, "10 min. The D-psicose" should be "10 minutes. The D-psicose". Response : The word “min” has not been changed as suggested for the consistency throughout the manuscript (line 178).
|
||
|
In line 180, "Twenty microliters of each sample were injected and the elution was performed" should be "Twenty microliters of each sample was injected, and the elution was performed". Response : The sentence has been revised as suggested (line 183).
|
||
|
In line 182, "50°C and pH 8.0 [12]" should be "50°C and pH 8.0". The sentence has not been corrected as suggested due to “[12]” is the citation for this method (line 185).
|
||
|
In line 183, "specific activity of DPEase (unit/mg) was calculated from DPEase activity (unit/ml) divided by concentration of protein (mg/mL)" should be "specific activity of DPEase (unit/mg) was calculated by dividing DPEase activity (unit/mL) by protein concentration (mg/mL)". Response : The sentence has been revised as suggested (line 186-187).
|
||
|
In line 189, "greater than 85%) in E. coli from previous report" should be "greater than 85%) in E. coli from a previous report". Response : The sentence has been revised as suggested (line 191).
|
||
|
In line 191, "Table 2. Most of the selected signal" should be "Table 2. Most selected signal". Response : The sentence has been revised as suggested (line 193).
|
||
|
In line 198, "Table 2 Cleavability" should be "Table 2: Cleavability". Response : The word has not been corrected as suggested due to the format of the journal (line 200).
|
||
|
In line 207, "Table 3 Localization" should be "Table 3: Localization". Response : The word has not been corrected as suggested due to the format of the journal (line 208).
|
||
|
In line 223, "reduction in DPEase solubility" should be "reduction in the solubility of DPEase". Response : The sentence has been revised as suggested (line 225).
|
||
|
In line 223, "solubility with a value of 0.89" should be "solubility, with a value of 0.89". Response : The sentence has been revised as suggested (line 226).
|
||
|
In line 228, "secretion efficiency for in vitro analysis, validating" should be "secretion efficiency for in vitro analysis, thereby validating". Response : The word thereby has been added as suggested (line 231).
|
||
|
In line 234, "DPEase. So, the DPEase" should be "DPEase; therefore, the DPEase". Response : The sentence has been revised as suggested (line 236).
|
||
|
In line 239, "the all four protein" should be "all four protein". Response : The sentence has been revised as suggested (line 242).
|
||
|
In line 244, "DPEase structure analysis" should be "DPEase structure analysis.". Response : The word has not been corrected as suggested due to the format of the journal (line 234).
|
||
|
In line 249, "BL21(DE3), T7 expression host" should be "BL21(DE3), a T7 expression host". Response : The sentence has been revised as suggested (line 253).
|
||
|
In line 251, "PCR, as shown in the Figure S2" should be "PCR, as shown in Figure S2". Response : The sentence has been revised as suggested (line 254).
|
||
|
In line 254, "with N- terminal selected" should be "with N-terminal selected". Response : The word has not been corrected as suggested due to the format of the journal (line 250-251).
|
||
|
In line 273, "under control of T7 promoter" should be "under the control of the T7 promoter". Response : The sentence has been revised as suggested (line 275).
|
||
|
In line 276, "from various compartment activity levels" should be "from various compartments' activity levels". Response : The sentence has been revised as suggested (line 277). |
||
|
In line 278, "samples. The highest" should be "samples, the highest". Response : The sentence has not been corrected as suggested due to more clarity with the separated sentence (line 281).
|
||
|
In line 282, "Psicose, respectively. A similar result" should be "Psicose, respectively; a similar result". Response : The sentence has not been corrected as suggested due to more clarity with the separated sentence (line 284).
|
||
|
In line 285, "samples. However, the enzyme" should be "samples; however, the enzyme". Response : The sentence has not been corrected as suggested due to more clarity with the separated sentence (line 288).
|
||
|
In line 287, "fractions were too low to be detected" should be "fractions were too low to detect". Response : The sentence has been revised as suggested (line 289). |
||
